# Gender Differences in the Association between Frailty, Cognitive Impairment, and Self-Care Behaviors Among Older Adults with Atrial Fibrillation

**DOI:** 10.3390/ijerph16132387

**Published:** 2019-07-05

**Authors:** Youn-Jung Son, Kyounghoon Lee, Bo-Hwan Kim

**Affiliations:** 1Red Cross College of Nursing, Chung-Ang University, Seoul 06974, Korea; 2College of Medicine, Division of Cardiology, Gachon University, Incheon 21565, Korea; 3College of Nursing, Gachon University, Incheon 21936, Korea

**Keywords:** atrial fibrillation, gender, frailty, cognition, self-care behaviors

## Abstract

Atrial fibrillation (AF), common in older adults, increases the risk of heart failure, stroke, and all-cause mortality. Self-care behaviors help avoid adverse events in older patients with AF. However, while frailty and cognitive impairment can contribute to poor self-care behaviors, few studies have explored these relationships in older adults with AF. This cross-sectional study aims to determine associations between frailty, cognitive impairment, and self-care behaviors among older adults with AF by gender. A total of 298 patients with AF aged 60 and over were assessed with a self-reported questionnaire consisting of the Korean version of the FRAIL scale, modified mini-mental state examination, and self-care scale for AF. Prevalence of frailty and prefrailty in men and women was around 11% and 48.4% and 28% and 47.4%, respectively. According to the hierarchical linear regression analysis, in men, prefrailty (β = −2.874, *p* = 0.013) and frailty (β = −7.698, *p* < 0.001) were associated with self-care behaviors; in women, frailty (β = −5.476, *p* = 0.003), and cognitive impairment (β = −3.350, *p* = 0.044) were associated with self-care behaviors. Developing individualized care plans will require periodic screening of older patients with AF to determine their frailty status and cognitive function.

## 1. Introduction

Atrial fibrillation (AF) is a common arrhythmia that increases the risk of morbidity and mortality in older men and women [1,2]. The progressive aging of the population has led to a noticeable increase in the morbidity of older adults with AF. Patients with AF are at a risk of mortality that is four times higher than the general population in South Korea [3]. Moreover, older adults with AF are exposed to its most severe complications, such as stroke [3,4], heart failure [4], and cognitive decline and dementia [5]. AF is one of the cardiovascular risk factors for dementia [6]. Although the mechanisms between AF and dementia are not fully understood, they may include: cerebral hypo-perfusion, inflammation, cerebral microbleeds, and recurrent silent cerebral ischemia [6,7]. These conditions can be more frequent in the AF population than among other cardiovascular patients [1]. Therefore, brain hypo-perfusion could accelerate poor frailty and cognition, like dementia, in AF patients [7]. 

Recently, frailty has been utilized as a construct to explore the health status of older adults [8]. Frailty is defined as a decrease in physiologic reserve and an increase in insufficient physical activity due to illness or injury [9]. According to previous studies, the prevalence of frailty among patients with AF is on the rise [10], and age-related declines in physical performance have accelerated by approximately four years for those with AF compared to those without AF [11]. In addition, among older adults admitted to the hospital, AF was four times more strongly associated with frailty after adjusting for confounding variables, such as age, sex, hypertension, diabetes, stroke, myocardial infarction, and heart failure [12]. Even though frailty is common in older patients with AF, Villani et al. [13] reported that it is not possible to draw any firm conclusions regarding the association between frailty and AF. More studies on the relationship between frailty and aging among patients with AF are needed to improve our understanding of the potential burden of frailty on the intricacies of long-term care, such as self-care behaviors. 

Self-care behaviors are crucial for the prevention and management of heart failure or stroke among patients with AF [14]. According to a systematic review, self-care for oral anticoagulation in AF was associated with a significant reduction in thromboembolic events [15]. Therefore, optimal care is the goal for older patients with AF [14,16]. Self-care behaviors may differ by sex, age, frailty status, and cognitive function [17,18,19]. There is a paucity of research to guide the development of strategies for promoting effective self-care behaviors to control symptoms, reduce complications, limit progression to permanent AF, and promote health in patients with AF [20]. Frailty in patients with AF might affect physical and cognitive function [11] and can lead to a restriction of self-care behaviors [19]. Although research has shown that frailty increases with age and that women are more likely to be frail [9,12], gender differences in the relationship between frailty and self-care behaviors in patients with AF have not been considered. 

Cognitive function may also affect long-term self-care behaviors in older populations [17,21]. Particularly, patients with AF with poor cognitive function may be unable to manage their self-care behaviors. Unfortunately, the relationship between cognitive function and self-care behaviors in patients with AF has not been studied. Importantly, frailty might be related to cognitive function [22,23].

In this study, we sought to find gender difference as a non-modifiable factor in AF patients. According to several studies, women with cardiac conditions are more likely to experience psychological distress, have poor functional status, and need more social support than men [18,24]. However, Dellafiore et al. [25] reported that men with chronic heart failure had more than quadruple the risk of poor self-care than women, while about 60 percent of men were more likely to have adequate self-care confidence than women, paradoxically. Therefore, in this study we assessed the effect of gender on associations between frailty, cognitive impairment, and self-care behaviors among older adults with AF.

To date, there are few studies concerning the impact of frailty and cognitive function on self-care behaviors in older patients with AF. Identifying the gender differences in these three variables is necessary for understanding and reducing inequalities in healthcare delivery [26]. Accordingly, this study investigated gender differences in the impact of frailty and cognitive impairment on self-care behaviors in older patients with AF in South Korea.

## 2. Materials and Methods

### 2.1. Study Design and Sample

We used a cross-sectional, correlational study design for the current study. Research associates recruited patients with AF who visited the outpatient clinics of the university-affiliated hospital in Incheon city in South Korea from February to August 2018 for their routine cardiology follow-up. 

Patients were eligible for the study if they (1) had received physician-confirmed diagnoses of AF and had been taking anti-thrombin medications for at least six months prior to the study, (2) were aged 60 or above, and (3) could speak Korean. The exclusion criteria were transient AF, severe diseases (cancer, chronic respiratory disease, terminal heart failure, severe dementia, depression, and psychiatric disorders); a history of cognitive problems, including mild cognitive impairment and dementia; and hearing impairment.

The sample size calculated for multiple regression analysis was 294, with an alpha of 0.05, power of 0.95, and effect size of 0.10, and the number of tested predicted factors and the total number of factors were set at 15 in G*Power 3.1.9.4 [27]. A total of 320 patients with AF were invited to participate; 15 patients did not meet all the inclusion criteria, and 7 patients declined to participate. Therefore, a total of 298 patients was sufficient for data analysis in this study. 

### 2.2. Measurements

#### 2.2.1. Sociodemographic and Clinical Characteristics

We obtained sociodemographic characteristics, including age, educational level, family type, job, monthly income, and body mass index. We reviewed medical records to acquire clinical characteristics (year of diagnosis of AF; type of AF; CHA_2_DS_2_-VASc score; hypertension, abnormal renal and liver function, stroke, bleeding, labile international normalized ratio (INR), older adults, and drugs or alcohol (HAS-BLED) score; comorbidities; medication; and lab data such as hemoglobin, hematocrit, prothrombin, partial thromboplastin time, and INRs). Labile INRs refer to unstable/high INRs or poor time in the therapeutic range [28]. The CHA_2_DS_2_-VASc score was calculated to evaluate stroke risk, and the HAS-BLED score was calculated as a measure of baseline bleeding risk [28,29]. Each clinical score was validated in previous studies [30,31]. 

#### 2.2.2. Frailty

Frailty was measured using the Korean version of the Fatigue, Resistance, Ambulation, Illnesses, and Loss of weight (FRAIL) scale [32], which was based on the original English FRAIL scale [33]. The FRAIL scale consists of five domains: fatigue (response of “all the time” or “most of the time”), resistance (ability to climb stairs), ambulation (ability to walk a certain distance), number of illnesses (five or more self-reported illnesses out of a total of 11), and loss of weight (more than 5% in the past year). The scores of the Korean version of the FRAIL scale range from 0–5 (i.e., 1 point for each added component; 0 = best to 5 = worst) and represent frail (3–5), prefrail (1–2), and robust (0) status. Patients’ degrees of frailty were judged according to the above scoring criteria.

#### 2.2.3. Cognitive Function

Cognitive function was measured using the Korean version of the Modified Mini-Mental State Examination (3MS) [34,35]. The 3MS is a brief cognitive test, including attention, orientation to time, memory, calculation, and language. The 3MS score ranges from 0 to 100 points, with higher scores indicating better cognitive function. Cognitive impairment is defined as a score of less than 72 in the South Korean older adult population without dementia [35]. Cronbach’s α in the current study was 0.84.

#### 2.2.4. Self-Care Behaviors 

Self-care behaviors were assessed with the self-care scale for patients with AF [36]. This scale consists of self-care resources (five items), self-care knowledge (four items), and self-care actions (five items). Responses were rated on a five-point Likert scale ranging from 1 to 5. The total score is calculated by adding the ratings from 14 to 70, with higher scores indicating better self-care behaviors. Cronbach’s α in the current study was 0.75.

### 2.3. Ethical Considerations and Data Collection

This study was approved by the Institutional Review Board of Gil Hospital (GBIRB2018-046). We obtained written informed consent from patients with AF in this study. The investigation conforms with the principles outlined in the Declaration of Helsinki. Before the research associates began recruiting participants, a dementia expert, who was not involved in patient assessments, re-evaluated prospective participants’ medical records. Then, the research nurse approached all eligible patients. Written, informed consent included details about the study aim, the voluntary nature of participation, and the confidentiality and anonymity of the information gathered. 

### 2.4. Data Analysis

We used the Kolmogorov–Smirnov test to assess the normality of the distribution of quantitative variables. Categorical variables were expressed as numbers and percentages. Continuous variables were expressed as mean and standard deviation (SD) in a normal distribution. For baseline variables, a one-way ANOVA, chi-square test, or Fisher’s exact test was used.

Hierarchical linear regressions were used to identify the impact of frailty and cognitive impairment on self-care behaviors by gender. In step 1, the covariates included the significant sociodemographic and clinical characteristics of both genders. In step 2, prefrailty, frailty, and cognitive impairment were added. Dummy variables were created for the independent variables with nominal or ordinal levels of measurement. We used tolerance and variance inflation factor (VIF) to check for multicollinearity. A VIF value less than 10 and tolerance value greater than 0.1 were acceptable. Differences were considered statistically significant at *p* < 0.05, and all analyses were two-tailed. We analyzed the data using SPSS 23.0 (IBM-SPSS Inc., Chicago, IL, USA).

## 3. Results

### 3.1. Patients’ Sociodemographic and Clinical Characteristics

A total of 298 older adults with AF—61.7% (*n* = 184) men and 38.3% (*n* = 114) women—participated in the study. The mean age was 72.11 ± 7.51 years. The prevalence of frailty among men and women was 11% and 28%, respectively. The prevalence of prefrailty among men and women was 48.4% and 47.4%, respectively. 

Frail men were older; less educated; earned a lower monthly income; and had higher stroke risk, bleeding risk, prevalence of hypertension, rate of heart failure, and rate of renal failure than robust or prefrail patients with AF. Additionally, frail women were older, were more likely to live alone, and had a higher bleeding risk than robust or prefrail patients with AF (Table 1 and Table 2). 

### 3.2. Differences in Cognitive Function and Self-Care Behaviors by Gender and Frailty Status

The prevalence of cognitive impairment differed significantly by frailty status only among female patients with AF (Table 3). Namely, a high proportion of frail, older adults, and female patients with AF experienced cognitive impairment (*p* = 0.001), which was not the case with male patients. 

Both frail men and women were less likely to perform self-care behaviors than robust and prefrail patients with AF (*p* < 0.001).

### 3.3. Frailty and Cognitive Impairment as Predictors of Self-Care Behaviors by Gender

In hierarchical linear regression analysis (Table 4 and Table 5), the predictors of self-care behaviors in male patients were prefrailty (β = −2.874, *p* = 0.013) and frailty (β = −7.698, *p* < 0.001). Regarding female patients, the predictors of self-care behaviors were frailty (β = −5.476, *p* = 0.003) and cognitive impairment (β = −3.350, *p* = 0.044). 

## 4. Discussion

To our knowledge, this study is the first to examine gender differences in the association between frailty, cognitive impairment, and self-care behaviors in the AF population. Our study found that the prevalence of frailty and prefrailty were 11% and 48.4% in men and 28% and 47.4% in women, respectively. One study reported that the prevalence of frailty and prefrailty were 48.2% and 18.6%, respectively, among hospitalized patients with AF in South Korea [10]. An Italian study reported that the prevalence of frailty and prefrailty in hospitalized patients with AF were 54% and 29%, respectively [37]. These variations from the results of our study could be due to sample-related differences; the two abovementioned studies [10,37] may have a higher prevalence of frailty than our study, because AF patients with high frailty are more likely to be hospitalized than to visit outpatient clinics [38]. The proportions of frailty and prefrailty in prior studies using the same tools of frailty measurement as ours were 17.5% and 58.3% in community-dwelling older people, which are similar to our findings. Thus, we need to consider frailty status based on patients’ illness trajectories.

In particular, the frailty of female patients with AF was higher than that of male patients; these results were in accordance with previous findings, particularly those of a study on community-dwelling older individuals in South Korea, which reported that frailty among women was greater than the age-matched frailty among men [8,38,39]. Interestingly, unlike previous studies [38], the prevalence of prefrailty in our study was higher in men than in women. Recent studies on AF have shown only the prevalence of frailty, without considering prefrailty, and have not researched gender differences [40,41]. Therefore, our study is significant in its stratification of frailty and prefrailty by gender in patients with AF.

The current study found the factors affecting men and women’s self-care behaviors differed. Prefrailty and frailty were important factors affecting men’s self-care behaviors, while frailty and cognitive impairment were the factors that most affected women’s self-care behaviors. It is interesting that impaired cognitive function negatively affected only women’s self-care behaviors. This result was associated with a previous study that found that women with mild cognitive impairment have greater longitudinal rates of cognitive and functional progression than men [42]. In one study concerning patients with heart failure [43], cognitive impairment was more prevalent among older women than men. Likewise, there have been reports that women have a higher prevalence of multimorbidity than men despite their longer life spans [44,45]. AF may be a marker of frailty syndrome and can be considered an indicator of an individual’s daily physical function [46,47]. Self-care behaviors play a fundamental role in health maintenance and the prevention and management of chronic illnesses like AF [14]. Our results imply that as modifiable factors for enhancing self-care behaviors, cognitive impairment, and frailty status are reversible clinical conditions if they are managed from the early stages of AF for older men and women. Thus, men’s and women’s unique situations should be considered in order to provide effective interventions for improving self-care behaviors in patients with AF. Furthermore, regularly researching the effects of prefrailty and frailty on self-care behaviors by gender is necessary. Finding the causes and effects between frailty, cognitive impairment, and self-care behaviors by gender will require close consideration of various sociocultural and biological aspects in future research [7,26].

This study has several limitations. First, it employed a cross-sectional design, which limits inferences regarding causality. Another limitation is that the sample is not representative of all South Korean patients with AF, thereby limiting the generalizability of the results. The personal interview data were based on self-reports; thus, recall bias and misunderstanding could have affected participants’ answers. Although cognitive impairment was a significant risk factor of self-care behaviors in female patients with a 95% CI -6.608 to -0.093 (*p* = 0.044), this finding denotes a small sample size that may not represent the population and statistically significant does not mean clinically significant. Moreover, the clinical utility of frailty and cognitive screening tools remains limited because assessments were conducted only on one clinical visit. Therefore, further research to validate these tools is required. Finally, we did not include possible risk factors, such as oxidative stress [48], cardiovascular reactivity [49], or risk of stroke [50], which can influence AF. More research is needed to identify the adverse health outcomes of risk factors on AF.

## 5. Conclusions

Our study shows that frailty and cognitive impairment are associated with self-care behaviors in both men and women with AF. Identifying gender differences in frailty status and cognitive function can help improve the self-care behaviors of older populations with comorbid conditions. Accordingly, the geriatric assessment of frailty and cognitive impairment should be considered when planning individualized care provisions for older adults with AF. It is necessary to develop self-management programs based on older patients’ frailty status and cognitive function. Future studies should be conducted with larger multicenter cohort sample sizes and long-term follow-ups in diverse healthcare settings.

## Figures and Tables

**Table 1 ijerph-16-02387-t001:** Sociodemographic characteristics by gender and frailty status (*n* = 298).

Characteristics	Total (*n* = 298)	Men (*n* = 184)				Women (*n* = 114)			
		Robust (*n* = 74)	Prefrail (*n* = 89)	Frail (*n* = 21)	*p*	Robust (*n* = 28)	Prefrail (*n* = 54)	Frail (*n* = 32)	*p*
	*n* (%) or M (SD)	*n* (%) or M (SD)	*n* (%) or M (SD)	*n* (%) or M (SD)		*n* (%) or M (SD)	*n* (%) or M (SD)	*n* (%) or M (SD)	
Age (years) *	72.11 (7.51)	69.47 (7.22)	71.78 (7.33) ^†^	74.05 (6.57) ^†‡^	0.019	71.00 (7.56)	73.65 (6.93)	76.28 (7.93) ^†^	0.024
60–69	119 (39.9)	42 (56.8)	35 (39.3)	4 (19.0)	0.016	15 (53.6)	16 (29.6)	7 (21.9)	0.037
70–79	122 (40.9)	25 (33.8)	38 (42.5)	12 (57.1)		6 (21.4)	27 (50.0)	14 (43.8)	
≥80	57 (19.1)	7 (9.5)	16 (18.0)	5 (23.8)		7 (25.0)	11 (20.4)	11 (25.4)	
Educational level									
Below middle school	169 (56.7)	30 (40.5)	35 (39.3)	15 (71.4)	0.023	19 (67.9)	43 (79.6)	27 (84.4)	0.283
Above high school	129 (43.3)	44 (59.5)	54 (60.7)	6 (28.6)		9 (32.1)	11 (20.4)	5 (15.6)	
Family type									
Live alone	57 (19.1)	6 (8.1)	12 (13.5)	5 (23.8)	0.147	5 (17.9)	14 (25.9)	15 (46.9)	0.034
Live with family	241 (80.9)	68 (91.9)	77 (86.5)	16 (76.2)		23 (82.1)	40 (74.1)	17 (53.1)	
Job (yes) *	72 (24.2)	30 (40.5)	22 (24.7)	4 (19.0)	0.047	5 (17.9)	10 (18.5)	1 (3.1)	0.086
Monthly income (KRW)									
<1,000,000	136 (45.6)	19 (25.7)	44 (49.4)	15 (71.4)	< 0.001	7 (25.0)	27 (50.0)	24 (75.0)	0.001
≥1,000,000	162 (54.4)	55 (74.3)	45 (50.6)	6 (28.6)		21 (75.0)	27 (50.0)	8 (25.0)	
BMI (kg/m^2^) **	24.38 (2.94)	24.37 (2.31)	24.53 (2.79)	23.37 (4.08)	0.232	24.14 (3.38)	24.62 (3.17)	24.47 (3.03)	0.811
Underweight (< 20)	18 (6.0)	3 (4.1)	3 (3.5)	3 (15.8)	0.132	2 (7.7)	4 (7.8)	3 (9.7)	0.990
Normal (20–25)	166 (55.7)	45 (60.8)	50 (58.1)	13 (68.4)		15 (57.7)	27 (52.9)	16 (51.6)	
Overweight (> 25)	103 (34.6)	26 (35.1)	33 (38.4)	3 (15.8)		9 (34.6)	20 (39.2)	12 (38.7)	

All the data were included in Table 1; * Fisher’s exact test, ** summation of percentage is not equal to 100% because of missing data, † significance compared to robust, ‡ significance compared to prefrail; *p* < 0.05; BMI, body mass index.

**Table 2 ijerph-16-02387-t002:** Clinical characteristics by gender and frailty status (*n* = 298).

Characteristics	Total (*N* = 298)	Men (*n* = 184)				Women (*n* = 114)			
		Robust (*n* = 74)	Prefrail (*n* = 89)	Frail (*n* = 21)	*p*	Robust (*n* = 28)	Prefrail (*n* = 54)	Frail (*n* = 32)	*p*
	*n* (%) or M (SD)	*n* (%) or M (SD)	*n* (%) or M (SD)	*n* (%) or M (SD)		*n* (%) or M (SD)	*n* (%) or M (SD)	*n* (%) or M (SD)	
Year after diagnosis of atrial fibrillation (AF)	7.96 (6.22)	9.29 (7.82)	7.63 (5.60)	7.02 (5.15)	0.184	7.36 (4.72)	8.13 (6.21)	6.70 (5.29)	0.513
Type of AF*									
Paroxysmal	184 (61.7)	43 (58.1)	49 (55.1)	12 (56.5)	0.980	19 (67.9)	41 (75.9)	20 (62.5)	0.412
Persistent	105 (35.2)	28 (37.8)	36 (40.4)	9 (42.9)		8 (28.6)	13 (24.1)	11 (34.4)	
Permanent	9 (3.0)	3 (4.1)	4 (4.5)	0 (0.0)		1 (3.6)	0 (0.0)	1 (3.1)	
CHA_2_DS_2_-VASc *									
Low and intermediate risk (0–1)	36 (12.1)	20 (27.0)	10 (11.2)	2 (9.5)	0.021	2 (7.1)	2 (3.7)	0 (0.0)	0.268
High stroke risk (≥ 2)	262 (87.9)	54 (73.0)	79 (88.8)	19 (90.5)		26 (92.9)	52 (96.3)	32 (100.0)	
HAS-BLED score *									
Low and intermediate risk (0–2)	71 (23.8)	28 (37.8)	18 (20.2)	3 (14.3)	0.017	10 (35.7)	7 (13.0)	5 (15.6)	0.039
High bleeding risk (≥ 3)	227 (76.2)	46 (62.2)	71 (79.8)	18 (85.7)		18 (64.3)	47 (87.0)	27 (84.4)	
Comorbidities									
Hypertension (yes) *	229 (76.8)	45 (60.8)	72 (80.9)	19 (90.5)	0.003	23 (82.1)	45 (83.3)	25 (78.1)	0.831
Diabetes mellitus (yes)	81 (27.2)	14 (18.9)	23 (25.8)	7 (33.3)	0.329	7 (25.0)	17 (31.5)	13 (40.6)	0.426
Coronary artery disease (yes)	92 (30.9)	23 (31.1)	24 (27.0)	9 (42.9)	0.359	8 (28.6)	16 (29.6)	12 (37.5)	0.694
Heart failure (yes)	115 (38.6)	19 (25.7)	35 (39.3)	14 (66.7)	0.002	9 (32.1)	20 (37.0)	18 (56.3)	0.115
Stroke (yes) *	62 (20.8)	12 (16.2)	21 (23.6)	7 (33.3)	0.205	5 (17.9)	14 (25.9)	3 (9.4)	0.187
Renal failure (yes) *	9 (3.0)	0 (0.0)	5 (5.6)	3 (14.3)	0.010	0 (0.0)	0 (0.0)	1 (3.1)	0.526
Medications									
Aspirin (yes)	131 (44.0)	44 (59.5)	37 (41.6)	9 (42.9)	0.063	12 (42.9)	16 (29.6)	13 (40.6)	0.402
Warfarin (yes)	104 (34.9)	23 (31.1)	34 (38.2)	6 (28.6)	0.536	14 (50.0)	20 (37.0)	7 (21.9)	0.075
NOAC (yes)	86 (28.9)	22 (29.7)	23 (25.8)	6 (28.6)	0.855	6 (21.4)	17 (31.5)	12 (37.5)	0.398
Lab data **									
Hb (mg/dL)	13.53 ± 1.83	14.47 ± 1.29	13.86 ± 1.86 ^†^	12.93 ± 2.48 ^†‡^	0.002	13.01 ± 1.44	12.65 ± 1.46	12.75 ± 2.00	0.651
Hct (%)	40.05 ± 5.26	42.14 ± 4.87	40.95 ± 5.20	38.64 ± 6.46 ^†^	0.026	38.80 ± 4.27	37.93 ± 4.28	38.31 ± 5.58	0.732
PT (sec)	18.44 ± 9.18	17.16 ± 7.45	19.82 ± 11.43	17.92 ± 9.18	0.233	19.41 ± 8.90	19.25 ± 8.34	15.63 ± 6.59	0.102
PTT (sec)	35.10 ± 10.55	34.88 ± 9.74	35.37 ± 9.26	33.91 ± 7.36	0.818	36.76 ± 15.40	35.22 ± 13.17	34.16 ± 8.52	0.765
INR	1.67 ± 0.85	1.54 ± 0.66	1.76 ± 1.01	1.79 ± 1.17	0.271	1.79 ± 0.78	1.70 ± 0.74	1.43 ± 0.63	0.134

All the data were included in Table 2; * Fisher’s exact test; ** summation of percentage is not equal to 100% due to missing data, † significance compared to robust, ‡ significance compared to prefrail; *p* < 0.05; NOAC, new oral anticoagulants; Hb, hemoglobin; Hct, hematocrit; PT, prothrombin time; PTT, partial thromboplastin time; INR, international normalized ratio.

**Table 3 ijerph-16-02387-t003:** Differences in cognitive function and self-care behaviors by gender and frailty (*N* = 298).

Variables	Total (*n* = 298)	Men (*n* = 184)				Women (*n* = 114)			
		Robust (*n* = 74)	Prefrail (*n* = 89)	Frail (*n* = 21)	*p*	Robust (*n* = 28)	Prefrail (*n* = 54)	Frail (*n* = 32)	*p*
	*n* (%) or M (SD)	*n* (%) or M (SD)	*n* (%) or M (SD)	*n* (%) or M (SD)		*n* (%) or M (SD)	*n* (%) or M (SD)	*n* (%) or M (SD)	
Cognitive function *									
Impairment (≤ 72)	45 (15.1)	4 (5.4)	8 (9.0)	4 (19.0)	0.143	3 (10.7)	10 (18.5)	16 (50.0)	0.001
Normal (> 73)	253 (84.9)	70 (94.6)	81 (91.0)	17 (81.0)		25 (89.3)	44 (81.5)	16 (50.0)	
Self-care behaviors	31.71 (7.06)	34.92 (6.29)	32.00 (6.86) ^†^	26.29 (5.69) ^†‡^	< 0.001	34.54 (6.57)	30.72 (6.84) ^†^	26.22 (5.22) ^†‡^	<0.001

* Fisher’s exact test, † significance compared to robust, and ‡ significance compared to prefrail; *p* < 0.05.

**Table 4 ijerph-16-02387-t004:** Hierarchical linear regression analysis of self-care behaviors in male patients with AF (*n* = 184).

Predictors	Step 1			Step 2		
	*Β*	*t*(*p*)	95% CI	*β*	*t* (*p*)	95% CI
Age (years)	−0.104	−1.121 (0.264)	−0.287 to 0.079	−0.059	−0.641 (0.523)	−0.240 to 0.122
Educational level (below middle school)	−1.680	−1.536 (0.126)	−3.838 to 0.479	−1.132	−1.057 (0.292)	−3.246 to 0.983
Job (no)	2.692	1.950 (0.053)	−0.035 to 5.419	2.274	1.720 (0.087)	−0.338 to 4.885
Monthly income (< 1,000,000 KRW)	−2.575	−2.089 (0.038)	−5.010 to −0.141	−1.336	−1.098 (0.274)	−3.740 to 1.068
CHA_2_Ds_2_-VASc (high stroke risk)	−0.243	−0.134 (0.893)	−3.827 to 3.340	−0.354	−0.203 (0.839)	−3.801 to 3.092
HAS-BLED (high bleeding risk)	0.136	0.078 (0.938)	−3.302 to 3.575	−0.268	−0.161 (0.872)	−3.556 to 3.020
Hypertension	−0.905	−0.595 (0.553)	−3.911 to 2.101	0.294	0.198 (0.843)	−2.634 to 3.221
Heart failure	−0.636	−0.567 (0.572)	−2.854 to 1.581	0.445	0.402 (0.688)	−1.744 to 2.634
Renal failure	−0.299	−0.092 (0.927)	−6.699 to 6.102	0.469	0.151 (0.880)	−5.677 to 6.616
Hb (g/dL)	0.729	1.210 (0.228)	−0.461 to 1.919	0.475	0.821 (0.413)	−.668 to 1.618
Hct (%)	−0.079	−0.407 (0.685)	−0.463 to 0.305	−0.089	−0.480 (0.632)	−0.457 to 0.278
Prefrail				−2.874	−2.523 (0.013)	−5.123 to −0.624
Frail				−7.698	−4.044 (< 0.001)	−11.458 to −3.939
Cognitive impairment				−2.528	−1.310 (0.192)	−6.341 to 1.285
	Adjusted R^2^ = 0.053, F (*p*) = 1.759 (0.054)			Adjusted R^2^ = 0.139, F (*p*) = 2.759 (0.001), R^2^ change = 0.086		

CI, confidence interval; Hb, hemoglobin; Hct, hematocrit; dummy variables: educational level (reference = over high school), job (reference = yes), CHA_2_Ds_2_-VASC (reference = low and intermediate risk), HAS-BLED (reference = low and intermediate risk), and prefrail and frail (reference = robust).

**Table 5 ijerph-16-02387-t005:** Hierarchical linear regression analysis of self-care behaviors in female patients with AF (*n* = 114).

Predictors	Step 1			Step 2		
	*Β*	*t*(*p*)	95% CI	*β*	*t* (*p*)	95% CI
Age (years)	−0.194	−2.122 (0.036)	−0.375 to −0.013	−0.054	−0.553 (0.581)	−0.248 to 0.140
Family type (live alone)	0.082	0.055 (0.956)	−2.868 to 3.031	0.757	0.533 (0.595)	−2.059 to 3.573
Monthly income (<1,000,000 KRW)	−3.513	−2.525 (0.013)	−6.272 to −0.754	−2.045	−1.494 (0.138)	−4.760 to 0.670
HAS-BLED (high bleeding risk)	1.010	0.588 (0.558)	−2.396 to 4.415	1.088	0.658 (0.512)	−2.191 to 4.368
Prefrail				−2.482	−1.605 (0.112)	−5.550 to 0.586
Frail				−5.476	−3.005 (0.003)	−9.090 to −1.862
Cognitive impairment				−3.350	−2.040 (0.044)	−6.608 to −0.093
	Adjusted R^2^ = 0.150, F (*p*) = 4.873 (< 0.001)			Adjusted R^2^ = 0.244, F (*p*) = 5.435 (< 0.001), R^2^ change = 0.094		

CI, confidence interval; dummy variables: family type (reference = live together), monthly income (reference = ≥ 1,000,000 KRW), HAS-BLED (reference = low and intermediate risk), and prefrail and frail (reference = robust).

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
