# Peer review of "Gender Differences in the Association between Frailty, Cognitive Impairment, and Self-Care Behaviors Among Older Adults with Atrial Fibrillation"

_ijerph, 2019, doi:10.3390/ijerph16132387_

Round 1
Reviewer 1 Report
The present paper aimed to explore the relationship between frailty, cognitive impairment and self-care in elderly patients wit atrial fibrillation, considering gender differences.
A few revisions are needed, as follows:
Materials and methods, line 78: How did you study cognitive impairment if you excluded patients with cognitive impairment? Please check again inclusion and exclusion criteria! Psychiatric disorders belong also to the exclusion criteria.
Table 2, Lab data: Hg is, probably, hemoglobin. Please replace with Hb because it looks like mercury. The same observation for table 4 (the footnote).
Page 2. Sociodemographic and clinical characteristics: Please mention that you included all the data in table 1.
Page 3, line 89: What is “Labile international normalized ratio”?
Page 10, lines 167-170. Please check again your statement about prefrailty and cognitive impairment considering that the 95%CI crosses the null hypothesis (1), despite p-value indicating significance. The confidence interval is the value you can rely upon over the p- value. Please consider this in Discussion and Conclusions.
Discussion should start with a paragraph including your own findings, emphasizing what is new in your study and no references should be included in those findings. After that you should compare your results with those of previous studies and provide also references.
Lines 206-209: Please rephrase the sentence starting with: “Considering the characteristics…”
Lines 2014-2015: Probably there is a limited number of studies,…not the studies are limited. Please rephrase and clarify!
Study limitations: Please also mention that you did not include oxidative stress in your study, considering that it influences atrial fibrillation (Gasparova I, Kubatka P, Opatrilova R, et al. Perspectives and challenges of antioxidant therapy for atrial fibrillation. Naunyn Schmiedebergs Arch Pharmacol. 2017;390(1):1-14. doi: 10.1007/s00210-016-1320-9), cardiovascular reactivity ( Myburgh C, Huisman HW, Mels CMC, et al. Cardiovascular reactivity and oxidative stress in young and older adults: the African-PREDICT and SABPA studies. Blood Press. 2019;:1-10. doi: 10.1080/08037051.2019.1609348) and risk of stroke (Wang J, Huang L, Cheng C, et al. Design, synthesis and biological evaluation of chalcone analogues with novel dual antioxidant mechanisms as potential anti-ischemic stroke agents. Acta Pharm Sin B. 2019;9(2):335-350. doi: 10.1016/j.apsb.2019.01.003).
Author Response
June 26, 2019
International Journal of Environmental Research and Public Health
Dear Reviewer 1,
We wish to thank you for your thoughtful comments and valuable feedback on the manuscript entitled “Gender Differences in the Association Between Frailty, Cognitive Impairment, and Self-Care Behaviors Among Older Adults with Atrial Fibrillation,” a revised version of which will be resubmitted to the International Journal of Environmental Research and Public Health. We have modified the manuscript according to your suggestions, rewriting and rephrasing sections to improve clarity, add further information, and explain the details of points that were previously vague. For your convenience, we have used red font for the revisions. We believe that the revised version of this paper will be of interest to the readership of the International Journal of Environmental Research and Public Health.
Thank you for your consideration. We look forward to hearing from you.
Response to Reviewer 1’s Comments
Point 1. In the Materials and methods section, line 78: How did you study cognitive impairment if you excluded patients with cognitive impairment? Please check again inclusion and exclusion criteria! Psychiatric disorders belong also to the exclusion criteria.
Response: We revised and added information to the section on inclusion and exclusion criteria. Please see the Materials and Methods section on pages 2-3, lines 88-93.
“Patients were eligible for the study if they 1) had received physician-confirmed diagnoses of AF and had been taking anti-thrombin medications for at least six months prior to the study, 2) were aged 60 or above, and 3) could speak Korean. The exclusion criteria were transient AF, severe diseases (cancer, chronic respiratory disease, terminal heart failure, severe dementia, depression, and psychiatric disorders); a history of cognitive problems, including mild cognitive impairment and dementia; and hearing impairment.”
Point 2. Table 2, Lab data: Hg is, probably, hemoglobin. Please replace with Hb because it looks like mercury. The same observation for table 4 (the footnote).
Response: We replaced Hg with Hb in Tables 2 and 4 (in the tables and footnotes). Please see the Results section, Table 2 on page 7 and on line 170 in the footnote and Table 4 on page 11 and on line 188 in the footnote.
Point 3. Page 2. Sociodemographic and clinical characteristics: Please mention that you included all the data in table 1.
Response: We stated that all the data were included in Table 1 in the Results section Table 1 footnote and that all the data were included in Table 2 in the Results section Table 2 footnote. Please see the Results section on pages 5 and 7, lines 166 and 169.
Point 4. Page 3, line 89: What is “Labile international normalized ratio”?
Response: We revised this section and explained that Labile international normalized ratios (Labile INRs) refer to unstable/high INRs or poor time in the therapeutic range. Please see 2.2.1. Sociodemographic and clinical characteristics of the Material and Methods section, on page 3, lines 106-107.
Point 5. Page 10, lines 187-189. Please check again your statement about prefrailty and cognitive impairment considering that the 95%CI crosses the null hypothesis (1), despite p-value indicating significance. The confidence interval is the value you can rely upon over the p- value. Please consider this in Discussion and Conclusions.
Response: (1) We checked our error statement about prefrailty and cognitive impairment. (2) We discussed it in the Discussion and Conclusions sections. Please see the Results section on page 10, lines 183-186 for response (1) and the Discussion section on page 13, lines 237-240 for response (2).
“In hierarchical linear regression analysis (Tables 4 and 5), the predictors of self-care behaviors in male patients were prefrailty (β = −2.874, p = .013) and frailty (β = −7.698, p < .001). Regarding female patients, the predictors of self-care behaviors were frailty (β = −5.476, p = .003) and cognitive impairment (β = −3.350, p = .044).”
“Although cognitive impairment was a significant risk factor of self-care behaviors in female patients with a 95% CI -6.608 to -0.093 (p=0.44), this finding denotes a small sample size that may not represent the population and statistically significant does not mean clinically significant. “
Point 6. Discussion should start with a paragraph including your own findings, emphasizing what is new in your study and no references should be included in those findings. After that you should compare your results with those of previous studies and provide also references.
Response: We incorporated your suggestion that the Discussion section start with a paragraph concerning our findings without references. Please see the Discussion section on page 13, lines 194-197.
“To our knowledge, this study is the first to examine gender differences in the association between frailty, cognitive impairment, and self-care behaviors in the AF population. Our study found that the prevalence of frailty and prefrailty were 11% and 48.4% in men and 28% and 47.4% in women, respectively.”
Point 7. Lines 206-209: Please rephrase the sentence starting with: “Considering the characteristics…”
Response: At Reviewer 2’s request, we revised and shortened the Discussion section by at least a third, without altering the message. In the Results section, we deleted the sentence that began “Considering the characteristics…” Please see the Discussion section, on page 13, lines 214-232.
Point 8. Lines 234-235: Probably there is a limited number of studies,…not the studies are limited. Please rephrase and clarify!
Response: We rephrased and clarified this sentence on the limited number of studies in the Discussion section. Please see the Discussion section, on page 13, line 233, which begins “This study has several limitations…”
Point 9. Study limitations: Please also mention that you did not include oxidative stress in your study, considering that it influences atrial fibrillation (Gasparova I, Kubatka P, Opatrilova R, et al. Perspectives and challenges of antioxidant therapy for atrial fibrillation. Naunyn Schmiedebergs Arch Pharmacol. 2017;390(1):1-14. doi: 10.1007/s00210-016-1320-9), cardiovascular reactivity ( Myburgh C, Huisman HW, Mels CMC, et al. Cardiovascular reactivity and oxidative stress in young and older adults: the African-PREDICT and SABPA studies. Blood Press. 2019;:1-10. doi: 10.1080/08037051.2019.1609348) and risk of stroke (Wang J, Huang L, Cheng C, et al. Design, synthesis and biological evaluation of chalcone analogues with novel dual antioxidant mechanisms as potential anti-ischemic stroke agents. Acta Pharm Sin B. 2019;9(2):335-350. doi: 10.1016/j.apsb.2019.01.003).
Response: We added your valuable suggestions to a paragraph on this study’s limitations in the Discussion section. Please see the Discussion section, on pages 13-14, lines 242-244.
“Finally, we did not include possible risk factors, such as oxidative stress [48], cardiovascular reactivity [49], or risk of stroke [50], which can influence AF. More research is needed to identify the adverse health outcomes of risk factors on AF.”

Reviewer 2 Report
In this cross-sectional study, Son and colleagues investigated the potential relationship between frailty, cognitive impairment and self-care behaviors among older adults with atrial fibrillation (AF) according to gender. The authors showed that both in older men and women with AF, frailty and/or cognitive impairment were associated with self-care behaviors. Thus, the authors suggest that periodical screening for frailty status and cognitive function is required to individualize at best care plans for those patients. Results are interesting. Nevertheless, I have some concerns, which need to be discussed. Please consider the following comments.
1. As far as I know, patients are considered as elderly in the studies from the age of 75 or 80. Please justify why you consider patients from the age of 60.
2. How and who diagnosed AF?
3. Did you screen and include consecutive patients during the study period or did you use a “convenience sample”? Please add a flowchart, indicating how many patients you screened, how many were eligible for study inclusion and how many were excluded and why.
4. There are too many tables and it would be very helpful for readers to summarize your key findings with figures.
5. The discussion is far too long and could be shortened by at least a third without altering the message.
Author Response
June 26, 2019
International Journal of Environmental Research and Public Health
Dear Reviewer 2,
We wish to thank you for your thoughtful comments and valuable feedback on the manuscript entitled “Gender Differences in the Association Between Frailty, Cognitive Impairment, and Self-Care Behaviors Among Older Adults with Atrial Fibrillation,” a revised version of which will be resubmitted to the International Journal of Environmental Research and Public Health. We have modified the manuscript according to your suggestions, rewriting and rephrasing sections to improve clarity, add further information, and explain the details of points that were previously vague. For your convenience, we have used red font for the revisions. We believe that the revised version of this paper will be of interest to the readership of the International Journal of Environmental Research and Public Health.
Thank you for your consideration. We look forward to hearing from you.
Response to Reviewer 2’s Comments
Point 1. As far as I know, patients are considered as elderly in the studies from the age of 75 or 80. Please justify why you consider patients from the age of 60.
Response: In this study, we considered AF patients aged 60 and above to be elderly. This decision was based on Lee et al. (2017), which reported:
3.1. Incidence of AF
“...... The overall incidence rates per 10,000 person-years were 3.0, 6.0, 14.3, 33.7, and 67.4 per 10,000 person-years among those aged 30–39, 40–49, 50–59, 60–69, and 70–79 years, respectively (Fig. 1B)…”
3.2. Prevalence of AF
“…AF prevalence increased in both sexes throughout the study period. AF prevalence increased with older age, ranging from 0.03% among individuals aged 20–29 years to 4.16% among those aged ≥ 80 years in 2015. Among individuals aged ≥ 60 years, 2.28% had AF in 2015. The temporal trend in prevalence according to age is shown in Fig. 2B. A linear increase in AF prevalence was found from age 50–59 years to > 80 years…”
Figure 1. Incidence of atrial fibrillation between 2008 and 2015 (per 10,000 person-years). Figure A is the annual incidence of atrial fibrillation stratified according to sex. Figure B is the incidence of atrial fibrillation according to age group in each year.
Figure 2. Prevalence of atrial fibrillation between 2008 and 2015. Figure A is the annual prevalence of atrial fibrillation stratified according to sex. Figure b is the temporal trends of the prevalence of atrial fibrillation according to age group.
Reference data from Lee, S.R.; Choi, E.K.; Han, K.D.; Cha, M.J.; Oh, S. Trends in the incidence and prevalence of atrial fibrillation and estimated thromboembolic risk using the CHA2DS2-VASc score in the entire Korean population. Int J Cardiol. 2017, 236, 226-231. https://doi.org/10.1016/j.ijcard.2017.02.039.
Point 2. How and who diagnosed AF?
Response: In 2.1. Study Design and Sample, we added a statement on how and by whom AF was diagnosed. Please see the Materials and Methods section, on page 2-3, lines 85-90.
“… Research associates recruited patients with AF who visited the outpatient clinics of the university-affiliated hospital in Incheon city in South Korea from February to August 2018 for their routine cardiology follow-up.
Patients were eligible for the study if they 1) had received physician-confirmed diagnoses of AF and had been taking anti-thrombin medications for at least six months prior to the study, 2) were aged 60 or above, and 3) could speak Korean.”
Point 3. Did you screen and include consecutive patients during the study period or did you use a “convenience sample”? Please add a flowchart, indicating how many patients you screened, how many were eligible for study inclusion and how many were excluded and why.
Response: Thank your valuable suggestion. However, we tried to explain how we selected the sample population without adding a flowchart. Please see the Materials and Methods section, 2.1. Study Design and Sample, on page 3, lines 96-98.
“……A total of 320 patients with AF were invited to participate, 15 patients did not meet all the inclusion criteria, and 7 patients declined to participate. Therefore, a total of 298 patients was sufficient for data analysis in this study.”
Point 4. There are too many tables and it would be very helpful for readers to summarize your key findings with figures.
Response: We respect your valid opinion. However, we thought it over and decided that it was better to express the key findings in a table than in a figure.
Point 5. The discussion is far too long and could be shortened by at least a third without altering the message.
Response: We generally revised and shortened the Discussion section without altering its meaning. Please see the Discussion section on pages 13-14, lines 193-244.
“4. Discussion
To our knowledge, this study is the first to examine gender differences in the association between frailty, cognitive impairment, and self-care behaviors in the AF population. Our study found that …… Thus, we need to consider frailty status based on patients’ illness trajectories.
In particular, the frailty of female patients with AF was higher than that of male patients; these results were in accordance with previous findings, particularly those of a study on community-dwelling older individuals in South Korea, which reported that …… Therefore, our study is significant in its stratification of frailty and prefrailty by gender in patients with AF.
The current study found the factors affecting men and women’s self-care behaviors differed. Prefrailty and frailty were important factors …… Finding the causes and effects between frailty, cognitive impairment, and self-care behaviors by gender will require close consideration of various sociocultural and biological aspects in future research [7, 26].
This study has several limitations. First, it employed a cross-sectional design, which limits inferences regarding causality. …… Finally, we did not include possible risk factors, such as oxidative stress [48], cardiovascular reactivity [49], or risk of stroke [50], which can influence AF. More research is needed to identify the adverse health outcomes of risk factors on AF.”

Reviewer 3 Report
This manuscript is focused on a very interesting field, gender differences and self-care behaviors among older adults; however, this document has several weaknesses mainly in the objective of the contribution to public health. Therefore, it cannot be accepted for publication in its current form.
Major comments
1. Introduction. The study is not justified, the authors point out "while frailty and cognitive impairment can contribute to poor self-care behaviors, few studies have explored these relationships in elderly patients with AF". In this sense, frailty and cognitive impairment affects the self-care in elderly patients of all health problems, ¿what would be the difference with patients with atrial fibrillation? . On the other hand, the authors point out the possible differences of self-care relative to the gender, ¿what would be the explanation? In this regard, the socio-cultural context must be considered.
2. Results. It would be convenient for this study to carry out a logistic regression analysis and estimate the Odds ratio.
3. Discussion. The findings regarding the influence of frailty and cognitive impairment on self-care are obvious, ¿what would be the contribution of this paper ?. On the other hand, the authors must explain the differences observed by gender, considering the sociocultural aspects.
Author Response
June 26, 2019
International Journal of Environmental Research and Public Health
Dear Reviewer 3,
We wish to thank you for your thoughtful comments and valuable feedback on the manuscript entitled “Gender Differences in the Association Between Frailty, Cognitive Impairment, and Self-Care Behaviors Among Older Adults with Atrial Fibrillation,” a revised version of which will be resubmitted to the International Journal of Environmental Research and Public Health. We have modified the manuscript according to your suggestions, rewriting and rephrasing sections to improve clarity, add further information, and explain the details of points that were previously vague. For your convenience, we have used red font for the revisions. We believe that the revised version of this paper will be of interest to the readership of the International Journal of Environmental Research and Public Health.
Thank you for your consideration. We look forward to hearing from you.
Response to Reviewer 3’s Comments
Point 1. Introduction. The study is not justified, the authors point out "while frailty and cognitive impairment can contribute to poor self-care behaviors, few studies have explored these relationships in elderly patients with AF". In this sense, frailty and cognitive impairment affects the self-care in elderly patients of all health problems, ¿what would be the difference with patients with atrial fibrillation? . On the other hand, the authors point out the possible differences of self-care relative to the gender, ¿what would be the explanation? In this regard, the socio-cultural context must be considered.
Response: We revised the paper to explain how (1) frailty and cognitive impairment affect self-care in elderly patients with AF as opposed to older adults with other health problems. Moreover, we pointed out (2) possible differences in self-care as they relate to gender and whether those differences offer any explanations. Please see the Introduction section, on page 1, lies 37-41 for response (1) and page 2, lines 70-77 for response (2).
(1) “AF is one of the cardiovascular risk factors for dementia [6]. Although the mechanisms between AF and dementia are not fully understood, they may include: cerebral hypo-perfusion, inflammation, cerebral microbleeds, and recurrent silent cerebral ischemia [6, 7]. These conditions can be more frequent in the AF population than among other cardiovascular patients [1]. Therefore, brain hypo-perfusion could accelerate poor frailty and cognition, like dementia, in AF patients [7].”
(2) “In this study, we seek to find gender difference as a non-modifiable factor in AF patients. According to several studies, women with cardiac conditions are more likely to experience psychological distress, have poor functional status, and need more social support than men [18, 24]. However, Dellafiore et al. [25] reported that men with chronic heart failure had more than quadruple the risk of poor self-care than women, while about 60 percent of men were more likely to have adequate self-care confidence than women, paradoxically. Therefore, in this study we assessed the effect of gender on associations between frailty, cognitive impairment, and self-care behaviors among older adults with AF.”
Point 2. Results. It would be convenient for this study to carry out a logistic regression analysis and estimate the Odds ratio.
Response: We appreciate your opinion. However, we thought it over and decided that it would be better to express it as a hierarchical linear regression than a logistic regression. “Self-care behaviors” as dependent variables are used as continuous variables in this study.
Point 3. Discussion. The findings regarding the influence of frailty and cognitive impairment on self-care are obvious, ¿what would be the contribution of this paper ? On the other hand, the authors must explain the differences observed by gender, considering the sociocultural aspects.
Response: We revised and explained the contribution of this paper and the differences observed by gender, considering the sociocultural aspects in the Discussion section. Please see the Discussion section on page 13, lines 221-232.
“…... Likewise, there have been reports that women have a higher prevalence of multimorbidity than men despite their longer life spans [44, 45]. AF may be a marker of frailty syndrome and can be considered an indicator of an individual’s daily physical function [46, 47]. Self-care behaviors play a fundamental role in health maintenance and the prevention and management of chronic illnesses like AF [14]. Our results imply that as modifiable factors for enhancing self-care behaviors, cognitive impairment and frailty status are reversible clinical conditions if they are managed from the early stages of AF for older men and women. Thus, men’s and women’s unique situations should be considered in order to provide effective interventions for improving self-care behaviors in patients with AF. Furthermore, regularly researching the effects of prefrailty and frailty on self-care behaviors by gender is necessary. Finding the causes and effects between frailty, cognitive impairment, and self-care behaviors by gender will require close consideration of various sociocultural and biological aspects in future research [7, 26].”

Round 2
Reviewer 1 Report
The present study aimed to explore the relationship between frailty, cognitive impairment and self-care in elderly patients with atrial fibrillation, considering gender differences.
Please accept the manuscript considering that it was significantly improved considering suggestions of the reviewers.
Reviewer 2 Report
The authors have well improved their manuscript. I have no further comments. Congratulations for this work.